# ANTI-DISTILLATION: IMPROVING REPRODUCIBILITY OF DEEP NETWORKS

## ABSTRACT

Deep networks have been revolutionary in improving performance of machine learning and artificial intelligence systems. Their high prediction accuracy, however, comes at a price of *model irreproducibility* in very high levels that do not occur with classical linear models. Two models, even if they are supposedly identical, with identical architecture and identical trained parameter sets, and that are trained on the same set of training examples, while possibly providing identical average prediction accuracies, may predict very differently on individual, previously unseen, examples. *Prediction differences* may be as large as the order of magnitude of the predictions themselves. Ensembles have been shown to somewhat mitigate this behavior, but without an extra push, may not be utilizing their full potential. In this work, a novel approach, *Anti-Distillation*, is proposed to address irreproducibility in deep networks, where ensemble models are used to generate predictions. Anti-Distillation forces ensemble components away from one another by techniques like de-correlating their outputs over mini-batches of examples, forcing them to become even more different and more diverse. Doing so enhances the benefit of ensembles, making the final predictions more reproducible. Empirical results demonstrate substantial prediction difference reductions achieved by Anti-Distillation on benchmark and real datasets.

## 1 INTRODUCTION

In the last decade, deep networks provided revolutionary breakthroughs in machine learning, achieving capabilities that were not even imagined ten years ago, and penetrating every domain of our lives. They have been shown to be substantially superior to classical techniques that optimized linear models on convex objectives. With this success, however, comes a price; the price of *irreproducibility*, at high levels that were unseen before with classical models (Dusenberry et al., 2020). Training even the same exact model with identical parameters and architecture on the same set of training examples can produce very different models if trained more than once. Two such models can have equal average prediction accuracy on validation data, but predict very differently on individual examples. The problem can be as extreme as having a *Prediction Difference (PD)* that is of the same order of magnitude as the predictions themselves. Perhaps some applications can tolerate such differences, but imagine applications as medical ones, where for the same symptoms, one model would predict one disease, and the other model would predict another. While in some way, this mimics real life, where individuals make conclusions based on what they learned and what they know, or dependent on the order in which they learn different topics (see, e.g., Achille et al. (2017); Bengio et al. (2009)), this is definitely not a desired behavior. Overall, perhaps, predictions are better, but for the individual cases in which predictions are different, the consequences can be irreversible.

Deep models are usually trained on highly parallelized distributed systems. They normally are initialized randomly, and are expected to find a nonlinear solution that fits the data best, minimizing a non-convex loss objective. Applying determinism (see, e.g., Nagarajan et al. (2018)) to the order in which data is seen and/or updated may not be an option, especially in extremely large scale systems. Due to such systems, even if the models are initialized identically (to some identical pseudorandom set of initialization values), the trained model sees the training examples in some random order and the updates are applied also with some randomness. Due to the non-convex nature, different training instances of the same model on the same dataset, may still find and converge to different optima, which may be all equal in average accuracy, but very different for individual examples.

This problem becomes even more critical in re-enforcement and online (or mini-batch) learning, where the model needs to make predictions and decisions while it continues to train on new additional examples. The decisions the model makes also dictate what new data is seen. If two models diverge from one another on the same data, they can then make different future decisions, affecting what additional data is seen by the model. This can enhance the divergence of two supposedly identical models even more. Online Click-Through-Rate (CTR) prediction (see, e.g., McMahan et al. (2013)), where ads are shown based on the current state of the model and then the model updates based on user reactions to the shown ads, is an example where irreproducibility can lead to two models which diverge over time very largely from one another.

Using ensembles of models (see, e.g., Dietterich (2000); Koren (2009)) has become a very popular technique in machine learning, and also in deep networks. Ensembles can reduce prediction uncertainty as shown in Lakshminarayanan et al. (2017). Averaging the predictions of multiple deep networks, referred to as *ensemble components*, in an ensemble turns out to also help reduce prediction differences between two (or more) such ensembles. Because of irreproducibility, and especially if each component is initialized differently, each of the components diverges to focus on a different slice of the solution space. Thus the average prediction, which has a reduced variance, is more reproducible. One point to note, however, is that this approach can trade performance accuracy to improve reproducibility. If the number of operations (or the number of learned parameters) is a constraint, then, the components of the ensemble must have narrower layers than a single deep network applied to the same task if comparison is done where complexity and resources are kept equal to both systems. While ensembling on the narrower components has better accuracy than that of a single narrow component, the ensemble as whole may have inferior accuracy than a comparable single network with the same complexity (or number of parameters). This will happen in very large scale systems where the single components of the model are not significantly over-parameterized to the training data. (In smaller scales, it is possible that narrower models are still expressive enough for the specific problem, and adding more parameters is no longer helpful in improving model accuracy.)

*Distillation* (see Hinton et al. (2015), and also Lan et al. (2018)) is a technique that is gaining popularity in deep networks to transfer knowledge of a complex model or complex ensemble of models (Mosca & Magoulas, 2018) to a simple model in a way that would allow the simple model to exhibit similar or close performance to that of the complex model. This is particularly important to models deployed on small devices that do not have the capacity of larger systems. The simple *student* model uses the prediction of the complex *teacher* model as a label, to try and become more similar to the teacher, which is a stronger model with better predictions. Various methods as in Crowley et al. (2018); Gou et al. (2020); Kim et al. (2018); Muller et al. (2020); Mun et al. (2018); Tang et al. (2020) have been developed to extract the most from distillation.

**Contribution:** Unlike distillation methods, in this paper, we apply a technique of opposite nature, *Anti-Distillation (AD)*. Leveraging the benefit of ensembles, we try to make the components as different as possible, so together they capture a larger subset of the solution space as an ensemble. This is done by adding a (regularization) loss that forces different components to diverge from one another. This can increase the diversification of the ensemble. Various regularization losses can achieve the effect of diversifying the components. We focus on correlation and covariance losses. A de-correlation loss can be obtained by minimizing the square of the Frobenius norm of the off-diagonal terms of the correlation matrix of the predictions of the ensemble components. De-correlation loss can be applied over correlations estimated from the predictions of a mini-batch of training examples. De-correlation has been applied internally on neurons of hidden layers in neural networks for decades (see, e.g., Shamir et al. (1993)). It was used in attempts to simplify the representations of the networks, by pruning neurons whose contributions are correlated with others, or for reducing overfitting (Cogswell et al., 2015). Unlike these works, here we apply de-correlation on full model predictions of the components of the ensemble for the mere purpose of diversifying the predictions of these components, so that the overall averaged prediction of the ensemble captures a wider portion of the optimization parameter space, and by doing so, is more reproducible.

**Related Work:** Over-parameterization of deep networks has been studied in many references (see, e.g., Denil et al. (2013); Han et al. (2015) and many others). Deep networks can thus find multiple explanations to the same datasets (even when examples are randomized (Zhang et al., 2016)). As described above, such different explanations may yield non-identical individual predictions, despite even consistent identical average performance over validation sets.

Techniques, such as ensembles and distillation are becoming very popular in deep networks, mainly in an attempt to deploy smaller models to which better knowledge is distilled. Specifically, for improving deployed models' reproducibility, *co-distillation* was proposed in Anil et al. (2018) (see also Zhang et al. (2018)). Co-distillation distills information in all directions between two (or more) models. Each model learns from the others, where they all try to agree on some solution. Then, only one model needs to be deployed, representing the solution that was agreed on. The deployed model is more reproducible relative to a model of equal capacity, but because models attempt to agree on a solution which is between the optima they converged to, there is some degradation in prediction accuracy. This degradation can be compensated by making the individual models more complex. While only one of the individual models is deployed, all of them must train, leading, despite deployment resource savings, to increased use of training resources.

A different technique that can improve model reproducibility is *transfer learning* (see, e.g., Raffel et al. (2019) and references within). While the motivation for transfer learning is to leverage knowledge that was learned in one domain to another, one can also initialize parameters of a new model to the values given by a converged model which has already been trained for the same task. On one hand, this would guarantee that the model is more likely to converge to the same solution, but on the other, it may not let the model explore other possible solutions. Unlike ensembles, that smoothen the objective, this approach anchors the model to a solution on the original highly non-convex objective. Transferring only a partial set of parameters may not be sufficient, as the model must still adjust its other parameters to the transferred values. Hence, this method may not be sufficient when we want to train models to express new information and parameters.

A different line of work was focused more on measuring various effects in training deep networks. For example, in Shallue et al. (2018), the effect of data parallelism in training of deep networks was studied. Such parallelism does affect irreproducibility, at least indirectly. Subsequently to our work presented here, correlation between activation strengths and PD was studied in Chen et al. (2020).

**Paper Outline:** The remainder of the paper is organized as follows. In Section 2, we describe the ensemble system on which we apply anti-distillation. Section 3 outlines different metrics that can be used to measure prediction differences between deep models. Section 4 describes Anti-Distillation. Finally, in Section 5, experimental results are described.

## 2 SYSTEM DESCRIPTION

We consider a standard ensemble of deep networks as depicted in Fig. 1. Several (3 in the figure) deep networks (which can be shaped like towers or differently), each with its own inputs, that can be identical for all components or different, produces *logit* outputs. For binary labels, these are scalars, that are converted to probabilities via the *Sigmoid* function $\sigma(z) = 1/(1 + \exp(-z))$. For multi-label problems, these can be vectors, converted to probabilities by *Softmax*. (Other scores and target values can also be used.) For inference, the outputs of the components are averaged either in probability (Output 1) or in logit (Output 2). Averaging can be uniform, or can be done as a mixture of experts (Shazeer et al., 2017), with some gating mechanism. In training, gradients are normally not allowed to propagate from the ensemble output to each component, which trains on its own on the training objective loss, although some systems may allow such propagation. Supervised training is done with standard training objectives with labeled examples on some label loss, such as cross-entropy loss. The training label loss can be the same or different for each of the ensemble components, but inference generates predictions for each of the components, which are averaged as described. The components may be identical in structure and training hyper-parameters (as in Fig. 1), or may have different structures and/or different training hyper-parameters (such as learning rates, and even training algorithm). The components may be trained on the same data examples, or on different subsets of the training datasets, as long as there is a non-empty intersection training set, on which AD training loss can be applied. Inputs enter the bottom layers in Fig. 1 of each of the ensemble components. These can be any types of inputs, including embeddings of non-measurable features that are learned together with the rest of the link weights and biases that are learned by the network. The parameters of link weights, biases, and embeddings can be initialized randomly with some variance. Specifically, sparse embedding inputs can be randomly initialized with some variance. In order to generate diversity between the different components, the initialization of each component is different from the others. For Anti-distillation, we can control the variance level of

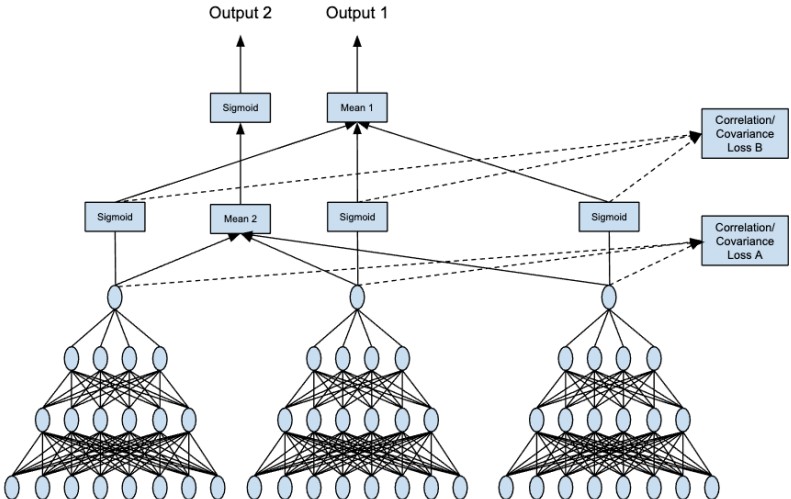

Figure 1: Ensemble of 3 components with Anti-Distillation: Each component is trained with its own loss. The ensemble prediction is an average of the predictions of the components (Output 1: in probability, Output 2: in logits). An additional AD loss is applied either on the logit values ($A$) or directly on the predictions ($B$).

the initialization of the input parameters (as well as other parameters). In addition to the training loss, an additional Anti-Distillation loss is added on the output (either in logits - $A$ in Fig. 1, or in probability - $B$). This loss attempts to push the components away from one another via de-correlation or another mechanism as will be described in Section 4. The AD loss can be applied on mini-batches of training examples, or continuously over training. While applied in training, AD loss does not change the inference, and has no effect on how a trained model is deployed for inference (in, e.g., a data center or a device).

## 3    PREDICTION DIFFERENCE

There are multiple approaches to measure *Prediction Difference (PD)* between trained models which perform inference over some validation dataset. PD can be a function of the model, its architecture, its parameters, the training algorithm used, the initialization, the random effects in the training system, and the dataset itself (both training and validation). We will measure PD on models that are supposed to be identical. Such models are defined with the same architecture, the same learned parameters, learning algorithm, and hyper-parameters, and are trained on the same training dataset. In some settings, we can assume that all the models in the set on which we measure PD are initialized identically. With pseudo-random initialization, the same sequence of pseudo-random values is used to initialize all parameters in any model, including embedding inputs and link weights and biases. Thus for any pair in the set, any learned parameter is initialized to the same value for the first and second models in the pair. Randomness leading to irreproducibility is thus the result of random effects on training and updates.

PD can be measured on the prediction score for regression and for classification (logit in the binary classification case) or the actual prediction values on the predicted labels in classification. We focus on the classification setup, and choose to measure PD on label predictions, although, this choice does not affect any of the conclusions. One can measure basic PD as some distance norm between a model's prediction and some baseline. Since in real systems, we do not have the "true" value of a prediction even if a label is observed, this value must be substituted by a function of the predictions on the given example in the set of models which we compare. Also, by definition, the metric we want to measure is the difference in prediction values among a set of models, so for such a difference the "true" probability does not play a direct role. In systems in which many multiple models can be trained, one can measure the PD for a given example relative to its expected prediction over all models. Let $M$ denote the number of different models, and $N$ the number of validation examples. Let $P_{n,m}$ be the distribution over labels predicted by model $m$ for example $n$, where $P_{n,m}(\ell)$ is the

probability predicted for label $\ell$. Let $P_n \triangleq \sum_m P_{n,m}/M$ be the averaged distribution over labels over all $M$ models. Then, the PD, $\Delta_p$, can be defined as the average $L_p$ norm of the difference between the distribution predicted by any of the models in the set and the expected distribution

$$\Delta_p = \frac{1}{N} \sum_{n=1}^{N} \cdot \frac{1}{M} \sum_{m=1}^{M} \|P_{n,m} - P_n\|_p = \frac{1}{N} \sum_{n=1}^{N} \cdot \frac{1}{M} \sum_{m=1}^{M} \cdot \left[ \sum_{\ell} |P_{n,m}(\ell) - P_n(\ell)|^p \right]^{1/p}. \quad (1)$$

Since we consider probabilities, $p = 1$ is reasonable. One can also weight the differences by $P_n(\ell)$ so that unlikely labels do not influence the measure. However, this is already achieved to some extent by measuring prediction differences on prediction probabilities instead of on the raw (logit for the binary case) scores. An alternative approach would be to use the KL-divergence between the predicted label distributions, or, since KL divergence is asymmetric, use the Jensen-Shannon divergence (see, e.g. Gutman (1989)). However, for small deviations, using Pinsker's inequality (Cover & Thomas, 2006), one can show that the KL divergence will be quadratic in the difference, and for small label predictions it will be inflated by the reciprocal of the lower label probabilities.

In practical very large scale problems, where training costs are high and resource availability is low, it may suffice to train a small set $M$ of models to have a notion of the PD behavior of a given model on a specific dataset. The definition in (1) generalizes to small values of $M$, unlike estimating, e.g., the prediction standard deviation among $M$ models. Specifically, for binary classification or regression problems, when one can effort training only $M = 2$ models, $\Delta_1 = 1/N \cdot \sum_n |P_{n,1}(1) - P_{n,2}(1)|$, where 1 is the positive label of interest.

In some applications, a more relevant bit of information is what fraction of the prediction differs between a model and a baseline. For example, in CTR prediction, where true CTRs are usually probabilities much smaller than $0.5$, the absolute differences may not be as informative. For $L_1$, we can thus define *relative PD*, $\Delta_1^r$, by normalizing the summand in (1) by $P_n(\ell)$. In binary problems, since there is only a single probability value to be estimated, that of a positive label, the summand can be normalized by $P_n(1)$ instead, leading to a slightly different metric denoted by $\tilde{\Delta}_1^r$. Instead, however, we can compute relative PD only on the label that is actually observed, replacing the sum on $\ell$ in (1) by the absolute difference for the true label, normalizing by $P_n(\ell_{\text{true}})$. We denote this metric by $\Delta_1^L$. If one has no access to $\ell_{\text{true}}$, $\tilde{\Delta}_1^r$ can be used instead of $\Delta_1^L$. Finally, in classification, one can use *Hamming PD*; $\Delta^H$, specifying the average of the labels predicted differently between model pairs. In the sequel we will show results focusing on $\Delta_1$, $\Delta_2$, $\tilde{\Delta}_1^r$ (binary case), $\Delta_1^L$, and $\Delta^H$.

## 4 ANTI-DISTILLATION

Consider an ensemble of $J$ internal components. Let $z_{j,t}$ be the signal on which AD loss is applied for the $j$th ensemble component on the $t$th training example. As noted in Fig. 1, $z_{j,t}$ can be either the logit/score value at the top of the $j$th deep component, or a prediction output of that component. We describe the approach for a scalar prediction value, but the methodology generalizes to a vector of values, e.g., representing scores or probabilities for different label values in the multi-label case. In that case, loss is applied in a similar way for each component of $z_{j,t}$, which is an output for a different label. While Fig. 1 restricts the signal to the prediction of the component, it is not necessarily the only point in which we can apply AD. The loss can be applied only on a subset network of the $j$th component. If the component consists of more complicated models combining linear components with deep ones, $z_{j,t}$ can also be only the output of a deep network part or of some combination of the elements of the component model. We now restrict the discussion to a single mini batch of training examples. Instead of using $z_{j,t}$, with some abuse of notation, let $\tau$; $\tau = 1, 2, \ldots, k$, be the example index in a mini-batch of $k$ examples, and let $z_{j,\tau}$ denote the prediction of the $j$th ensemble component on the $\tau$th example in the mini-batch.

Cogswell et al. (2015) used de-covariance loss on neurons of a hidden layer to push them apart. Here, we use either de-correlation or de-covariance on the predictions $z_{j,\tau}$. Now, let $C_z$ be the estimated correlation matrix between the $J$ prediction vectors in the mini-batch. The $i, j$ component of $C_z$ can be estimated by

$$C_{z;i,j} = \frac{1}{k} \sum_{\tau=1}^{k} z_{i,\tau} z_{j,\tau}. \quad (2)$$

Similarly, we can estimate the expected prediction score over the mini-batch of the $i$th ensemble component by $\mu_i \triangleq 1/k \cdot \sum_\tau z_{i,\tau}$, and estimate the elements of the *batch* covariance matrix $\Sigma_z$ by

$$\Sigma_{z;i,j} = \frac{1}{k} \sum_{\tau=1}^{k} (z_{i,\tau} - \mu_i)(z_{j,\tau} - \mu_j). \tag{3}$$

Instead, let $\tilde{\mu}_\tau \triangleq 1/J \cdot \sum_{i=1}^{J} z_{i,\tau}$ be the average ensemble prediction for example $\tau$ in the mini-batch. Then, we can compute the *residue correlation* between the components as

$$\tilde{C}_{z;i,j} = \frac{1}{k} \sum_{\tau=1}^{k} (z_{i,\tau} - \tilde{\mu}_\tau)(z_{j,\tau} - \tilde{\mu}_\tau). \tag{4}$$

(Normalizing by $k$ and not $k-1$ in (2)-(4) has no significant difference here, as batches can be large, and this term is absorbed in the learning rate parameters and the temperature of this loss relative to the main training loss, described below.) The AD loss is defined as

$$L_{\text{AD}}(C_z) = \frac{1}{2} \left[ \|C_z\|_F^2 - \|\text{diag}(C_z)\|_F^2 \right] \tag{5}$$

where $\|A\|_F$ denotes the Frobenius norm of the matrix $A$, and $\text{diag}(A)$ represents the diagonal matrix consisting of the diagonal elements of $A$. Hence, the correlation loss objective attempts to minimize the sum of squares of off-diagonal elements of the correlation matrix, thereby attempting to minimize the correlations between prediction vectors of the components of the ensemble. Similarly, we can replace $C_z$ in (5) by the covariance matrix $\Sigma_z$, or by the residue correlation matrix $\tilde{C}_z$. Correlation loss may be advantageous in most common situations where we want to de-correlate the full prediction signal, including the bias, and specifically if ensemble components are completely disjoint. If examples in a mini-batch share a subset of identical common features, and we want to emphasize de-correlations on the contributions of other features, which they do not share, using covariance over the batch is reasonable. This may be the case in sponsored advertising, where a mini-batch may include examples of impressions for some common query, where all examples share query features, but not ad features. In very large scale systems, it may be too costly to replicate all model parts in an ensemble, and ensemble components may share parts, such as embeddings, or partial embeddings. Models may also have components responsible to different parts of the signal, that may be shared. If shared components are linear and more reproducible, it may be more beneficial to use residue correlation loss, after subtracting the common contribution shared among components.

The AD loss should be weighted with some *temperature* $\lambda$ relative to the training (label) loss of the ensemble component. Thus the overall loss can be expressed as

$$L = L_{\text{label}} + \lambda L_{\text{AD}}. \tag{6}$$

Increasing $\lambda$ will, up to some point, provide smaller PD values between two or more AD ensemble models, but it may degrade the average prediction accuracy of the model. Such degradation can be compensated by a larger network. Tuning $\lambda$ can be used to generate favorable tradeoffs between accuracy and PD with minimal expansion of the model. Values too large, however, will diminish prediction effectiveness of the model. If AD loss is present with $\lambda$ small enough, random initiailization of model parameters (especially of parameter rich embeddings and layers closer to the input) with higher variance may improve PD without affecting accuracy.

The de-correlation loss can be further enhanced by applying techniques like Gram-Schmidt orthogonalization on the mini batch vectors $z_i$ before applying the AD de-correlation loss. This could ensure that the loss does not count correlated components more than once if the number of ensemble components is large. Similar or close effect, however, may be achievable by tuning the learning parameters and the temperature $\lambda$ of the AD loss.

In continuous online training regimes, it may be possible to extend the correlation estimates to be updated continuously over the batches of data observed, and provide a more global estimate of the correlation. This may be beneficial if the mini-batches are small, but may not give much benefit if mini-batches are large enough. Generally, a global estimate of the prediction correlation between the ensemble components may not give a current training view of the mini-batch correlations, as AD is attempting to lower it. It may thus dilute the effects of the AD loss attained by previous mini-batches, possibly overestimating the already lowered correlation at a given mini-batch.

Table 1: Anti Distillation on logit scores on MNIST with orrelation loss $L_{AD}(C_z)$.

| $\lambda$ | Log Loss | Acc | $\Delta_1$ | $\Delta_2$ | $\Delta_1^L$ | $\Delta^H$ |
|---|---|---|---|---|---|---|
| No Ad | 0.615 | 99.2% | 0.0777 | 0.0305 | 0.0183 | 0.0064 |
| 0.02 | 0.615 | 99.2% | 0.0040 | 0.0025 | 0.0052 | 0.0038 |
| 0.05 | 0.615 | 99.2% | 0.0037 | 0.0024 | 0.0051 | 0.0036 |
| 0.1 | 0.616 | 99.2% | 0.0036 | 0.0023 | 0.0051 | 0.0037 |
| 0.2 | 0.615 | 99.2% | 0.0036 | 0.0024 | 0.0052 | 0.0035 |
| 0.5 | 0.615 | 99.2% | 0.0036 | 0.0024 | 0.0052 | 0.0037 |
| 1 | 0.619 | 99.0% | 0.0063 | 0.0040 | 0.0081 | 0.0057 |

Table 2: Anti-Distillation on MNIST with correlation loss and varying ensemble size $J$.

| | **No Anti-Distillation** | | | | **Anti-Distillation** ($L_{AD}(C_z)$), $\lambda = 0.5$) | | | |
|---|---|---|---|---|---|---|---|---|
| $J$ | Acc | $\Delta_1$ | $\Delta_2$ | $\Delta_1^L$ | Acc | $\Delta_1$ | $\Delta_2$ | $\Delta_1^L$ |
| 2 | 99.2% | 0.0777 | 0.0305 | 0.0183 | 99.2% | 0.0036 | 0.0024 | 0.0052 |
| 4 | 99.2% | 0.0702 | 0.0273 | 0.0237 | 99.2% | 0.0018 | 0.0012 | 0.0037 |
| 6 | 99.2% | 0.0612 | 0.0236 | 0.0265 | 99.2% | 0.0012 | 0.0008 | 0.0029 |
| 8 | 99.1% | 0.0459 | 0.0177 | 0.0236 | 99.2% | 0.0009 | 0.0006 | 0.0024 |

Finally, other forms of loss can also be used, as long as they generate a similar effect of pushing predictions of components of the ensemble away from one another. A simple example is square loss that pushes the predictions away from one another. On a mini-batch, the loss can be expressed as

$$L'_{AD} = -\frac{1}{J^2 - J} \cdot \sum_{i=1}^{J} \cdot \sum_{j=1, j \neq i}^{J} \frac{1}{k} \sum_{\tau=1}^{k} (z_{i,\tau} - z_{j,\tau})^2. \tag{7}$$

The negative sign forces the predictions away from one another instead of towards each other. This loss tries to maximize the differences between predictions of the ensemble components on individual examples, whereas de-correlation loss can tolerate individual predictions that are similar as long as they are offset by other examples in the mini-batch. Alternatively, one can use $1/(|L'_{AD}| + \varepsilon)$ with some small $\varepsilon > 0$ to guarantee large gradients when the predictions are close to one another. Norms, other than $L_2$, can be used instead, as well as cross entropy losses. However, most experimental results suggest that de-correlation has been superior to such other losses, possibly because such losses are too restrictive on the individual examples. As such they may be pushing too much against the top level label training loss, whereas de-correlation may push in different directions on the batch vector. If the loss is applied on the batch, the network can still find different optima for each component that still generate predictions that are as good in average among the different ensemble components, but provide different individual predictions to some examples but not to all of them.

## 5 EXPERIMENTS

**MNIST:** We applied AD on the MNIST (LeCun, 1998). An ensemble component produces 10 *logit* outputs, on which softmax is applied to produce a label probability. Each component trains separately with cross entropy loss. AD loss (5) is applied to each of the 10 logit outputs of ensemble components (one loss for each label value). Inference on the test set only forward propagates, with no effect of the AD loss. Each model was trained over the 60,000 images in the MNIST training set, and ensemble prediction probabilities were averaged for each of the $N = 10,000$ images in the MNIST test set. Each configuration was trained $M = 20$ independent times.

Table 1 shows loss, accuracy and PD metrics on the MNIST test set of ensembles that consist of $J = 2$, 2 layer components each, with both layers of width 1200, trained over 150 epochs, using AdaGrad (Duchi et al., 2011) (learning rate 0.02, with accumulators initialized to 0.1), zero initialization of biases, and uniform initialization of the weights. Results are shown for different strengths $\lambda$ of AD

Table 3: Anti Distillation on real data with various losses.

| AD type | $\lambda$ | $s$ | Log Loss change % | Rank Loss change % | $\tilde{\Delta}_1^r$ |
|---|---|---|---|---|---|
| No AD | 0 | 1 | 0.000 | 0.000 | 0.0650 |
| $L_{AD}(C_z)$ | 0.5 | 1 | 0.379 | 0.073 | 0.0473 |
| | 1 | 1 | 0.903 | 0.126 | 0.0422 |
| | 5 | 1 | 4.660 | 0.342 | 0.0288 |
| | 0.5 | 5 | 0.379 | 0.071 | 0.0452 |
| | 1 | 5 | 0.903 | 0.128 | 0.0403 |
| | 5 | 5 | 4.656 | 0.324 | 0.0288 |
| | 0.5 | 10 | 0.381 | 0.078 | 0.0411 |
| | 1 | 10 | 0.907 | 0.143 | 0.0365 |
| | 5 | 10 | 4.660 | 0.349 | 0.0266 |
| $L_{AD}(\Sigma_z)$ | 1 | 1 | 0.018 | 0.011 | 0.0612 |
| $L'_{AD}$ | 1 | 1 | 0.069 | 0.120 | 0.0521 |

with correlation loss on logits. The results demonstrate that as soon as we apply AD, while accuracy and loss are not affected, all measures of PD improve substantially. The effect of the AD strength on PD here is not substantial, as long as AD is not applied too aggressively. Applying AD with too much strength degrades accuracy and PD. Similar behavior to Table 1 was obtained with covariance loss. Applying AD loss on probabilities did not improve PD.

In Table 2, the effect of the ensemble size $J$ is studied. Each component is a 2 layer network as above. Correlation loss uses $\lambda = 0.5$. The results show improvements to all PD metrics with no effect on accuracy as we increase the number of ensemble components. However, even with only two components, with AD, PD is better than 8 components without AD, and for every configuration, substantial improvements to PD are obtained with AD.

**Real Data:** AD was tested on a large real-world private dataset for ad Click-Through-Rate (CTR) prediction for sponsored advertisement. Training was applied on hundreds of billions of examples in a single pass over mini-batches of the data. *Progressive validation* methodology (Blum et al., 1999), standard in this domain (McMahan et al., 2013), measured validation performance. Results are reported for relative PD; $\tilde{\Delta}_1^r$, and logarithmic and ranking losses. We trained a network, consisting of 3 ensemble components with several sparse embedding input vectors representing informative input features, feeding into towers of narrowing layers, activated with ReLU. A single logit output is converted to a prediction of a positive label. The models were trained using AdaGrad with random truncated normal initialization with a standard variance normalized by the layer width. Input embeddings are trained with the network and were initialized similarly to hidden weights. We also studied the effect of scaling the input initialization variance by strength $s$. Each of the $M$ models compared used the same pseudorandom generation for initialization. Results show improved PD with increased $\lambda$, but with degradation in accuracy. PD also improves without effect to accuracy with increasing $s$, but improvements diminish with larger $\lambda$. Both covariance and difference losses improve PD less than correlation. Improvements are obtained applying PD on logits, but not on probabilities.

# 6 CONCLUSIONS

Irreproducibility is a problem when predicting on individual examples with deep networks. We presented various metrics to measure *Prediction Differences*. We proposed Anti-Distillation, a technique that leverages ensembles by encouraging their components to capture different optima in the objective space. AD improves reproducibility beyond ensembling, while maintaining the superior accuracy benefits of deep models. We studied various trade-offs with AD, and reported empirical results on benchmark and real datasets, demonstrating how AD can lower PD on validation data.

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
