# OpenReview forum: "Anti-Distillation: Improving Reproducibility of Deep Networks"
_ICLR.cc/2021/Conference — Reject_

### Official Review · AnonReviewer2 · 2020-10-24
**Confusing and difficult to read. Comparison with other work missing. Motivation not clear and misaligned with some content of the paper.**

**Rating:** 3
**Confidence:** 3

**Review:**

Summary:
The author proposes a method to train ensembles to have different prediction by using a correlation loss between the model's predictions. The authors show that their loss decreases the relative prediction difference between the models in the ensemble.

Strong points:
The correlation loss is a great idea to make model predictions as different as possible.

Weak points:
- The paper is difficult to read, and the line of reasoning is difficult to follow. It might be that I am just unfamiliar with the details of literature, but for me, it seems the authors are jumping back and forth between different interpretations and motivations.
- It is not entirely clear why the authors work on an advertisement dataset and why consistent predictions between ensembles are relevant.
- The experimental compare only against a baseline model, but the results are not grounded in the literature by comparing it to other methods.
- It is not entirely clear why de-correlation within an ensemble can be seen as "anti-distillation."

Recommendation (short):
This work's motivation seems to be misaligned with the content of the paper, and the experimental setup offers no comparison with similar work. I do not think this work reaches the standard of ICLR and would like to see it rejected.

Recommendation (long):
Please see the weak points.

Comments for authors:
Do not be discouraged by my recommendation. This work has some interesting results, but it currently just a little bit below scientific standards. The major flaws are the experiments are difficult to compare on their own without any comparison with other methods. The private(?) data does not help make it easy to understand the results, and the MNIST results are a bit toy-ish. I see related work does use the Criteo Display Ad Challenge dataset, which is also an advertisement dataset which you can compare with other methods. Beyond this, your motivation is initially about false positive/false negatives in the medical realm and their impact on people as a motivation for more robust methods. For me, it is not entirely clear why de-correlation helps and how the work can be seen as "anti-distillation." I can see the analogy to co-distillation, so for me, it looks like co-distillation with de-correlation loss, but why is this interesting? Furthermore, the writing is difficult to follow, and you do not seem to use a latex library for citations/figures, which makes it difficult to navigate the citations. I also see some obvious spelling and grammatical errors — please run your paper through a spellchecker like Grammarly. Some of the related work is strange: Why is your work related to the model pruning literature?

Overall, I would recommend working more on this paper, cleaning up all the major issues, and resubmitting it to another conference for more feedback and learning more from the process. If possible, it might also be useful for you to look for more experienced collaborators to improve writing and the overall experimental design process and the laying out of an argument based on those experiments.

---

### Official Review · AnonReviewer1 · 2020-10-24
**The motivation of improving reproducibility is not convinced.**

**Rating:** 3
**Confidence:** 3

**Review:**

This paper proposes the Anti-Distillation method to encourage prediction diversity in an ensemble model, in order to improve reproducibility. As I understand, the reproducibility defined in this paper refers to the prediction variance w.r.t. the random factors during training, e.g., SGD.

However, a trivial but complete reproducibility can be achieved by simply fixing the random seeds during training (without affecting model performance). Then why we would prefer the (incomplete) reproducibility induced by Anti-Distillation? If the reproducibility is the metric, then a single model or an ensemble model with fixed random seeds would trivially be the best model.

Besides, the experiments are done on MNIST and a private dataset. I suggest the authors evaluating their method on public datasets like CIFAR or ImageNet, where there are many existing baselines to compare the model performance.

---

### Official Review · AnonReviewer4 · 2020-10-27

**Rating:** 3
**Confidence:** 4

**Review:**

## Summary

The paper describes the problem of "model irreproducibility" - the fact that two models (with same architecture and initial weight values) trained on the same data (in different order) do not give the same prediction on the same data point at test time, while they do have the same accuracy on the test set.

The authors propose a method to quantify this effect using by looking at the absolute distance between the average prediction of an ensemble and predictions of elements of the ensemble.

The authors also propose a loss penalty to combat this effect called "anti-distillation". The penalty computes the correlation matrix between the average prediction of an ensemble and the prediction of an element of the ensemble and is trained to reduce the sum of squares of the off diagonal elements.

## Review

One of the motivations of the authors to quantify and attempt to reduce the effect of "model irreproducibility" is medical imaging, where one model would predict the current image contains a pathology but another one wouldn't. This is indeed a worrisome prospect of the use of ML in diagnostics.

This problem has however been widely studied in the Bayesian literature, and is an instance of model uncertainty. If we take for example a Gaussian Process and sample a function from its predictive posterior, then that function has some predictions for certain inputs, however if we sample another function it might be different. In fact, whenever we find a lot of variation in the predictions for the same input then we consider to have large uncertainty on that input.

Ensembles of neural networks allow for an effective way of quantifying uncertainty by looking at the entropy of the average prediction. If this entropy is high (meaning different classes where predicted by different ensemble elements, or the ensemble elements predict with low confidence) then the predictive uncertainty is high.

Using the concept of uncertainty, it's possible to signal to a medical practitioner that the model should not be trusted on this particular input. This approach is much preferable than if we attempt to hide this diversity, and instead focus all models on a single prediction. In fact: the diversity is important information that we can use! Avoiding it using anti-distillation, might give a false sense of security for the user of the system, but the underlying uncertainty is still there.

If the authors are interested in distilling an ensemble of models into a single model (for improved accuracy/uncertainty), which would be the outcome if the AD regularizer goes to zero, then additional comparisons need to be made for example with for example Malinin et al, 2019.

In conclusion, I think the authors look at the important problem applying ML in automated decision making, but the proposed approach should be reworked to consider the concept of uncertainty and is currently not a viable solution to the problem set out.

### References:

Gal, Yarin. "Uncertainty in deep learning." University of Cambridge 1.3 (2016).

Gal, Yarin, and Zoubin Ghahramani. "Dropout as a bayesian approximation: Representing model uncertainty in deep learning." international conference on machine learning. 2016.

Kendall, Alex, and Yarin Gal. "What uncertainties do we need in bayesian deep learning for computer vision?." Advances in neural information processing systems. 2017.

Lakshminarayanan, Balaji, Alexander Pritzel, and Charles Blundell. "Simple and scalable predictive uncertainty estimation using deep ensembles." Advances in neural information processing systems. 2017.

Malinin, Andrey, Bruno Mlodozeniec, and Mark Gales. "Ensemble distribution distillation." arXiv preprint arXiv:1905.00076 (2019).

Example medical applications:
Filos, Angelos, et al. "A Systematic Comparison of Bayesian Deep Learning Robustness in Diabetic Retinopathy Tasks." arXiv preprint arXiv:1912.10481 (2019).

## Notes

Table 1: "orrelation" -> "correlation"

MNIST results: difficult to interpret the deltas. You improved on reducing prediction diversity, it's not unreasonable that this should work given your objective, but why is it useful?

Why doesn't an ensemble of models improve the accuracy? On other datasets (such as cifar-10) that does happen and you might be able to measure more PD difference there.

Difficult to interpret "Real Data", not enough details on the dataset. "Training was applied on billions of examples in a single pass over mini-batches of data" - does this mean you trained using one epoch? Why does it matter for the CTR if models disagree on their prediction? Isn't the original data very sparse/noisy anyway? It's unclear to me why this dataset was chosen given the motivation in the introduction.

The paper is written with many tangents, where alternatives are suggested but never analysed and limited motivation is given for the current approach. For example, why don't you use an entropy penalty on the average prediction of the ensemble? The paper can benefit from increasing the clarity, motivation and additional discussion of the results.

---

### Official Review · AnonReviewer3 · 2020-10-30
**Anti-Distillation proposes a decorrelation loss for the predicts of ensemble networks to improve prediction quality**

**Rating:** 3
**Confidence:** 4

**Review:**

This `work applies the batch decorrelation loss proposed in  Reducing Overfitting in Deep Networks By Decorrelating Representations [Cogswell '15] to the final predictions an ensemble of neural networks in order to produce more diverse ensembles.

The Anti-Distillation approach proposed is a method for learning diverse models which may reduce overfitting and improve performance in real-world task. Outside of applying this loss to the logits or predictions from various networks it is unclear what is the main contribution of this work relative to related works. There are mentions of various potential types of AD (Anti-distillation loss) but no ablative studies and the loss used seems identical to prior work. The tasks which this approach is tested on, MNIST and private CTR data, show negligible performance change and the results aren't clearly explained. Please provide clear captions on the charts in future and perhaps test on a tasks where ensemble methods show significant performance improvements.

An approach of this style may very much be beneficial and useful for the community in setting where overfitting significantly impacts model performance. I'd encourage the authors to investigate more challenging domains, ColoredMNIST may be an appropriate alternative. As is the work should be arranged to more clearly present the contributions of the authors and the effects of the proposed method should be more clearly illustrated.

---

### Decision · Program_Chairs · 2021-01-07
**Final Decision**

**Decision:**

Reject

**Comment:**

The paper tries to argue the value of making ensembles more
reproducible through the use of a correlation loss to try to make
components as different as possible. The paper is tough to follow and
the high level motivation is unclear. As one of the reviewers points
out, don't ensembles provide an estimate of uncertainty and
calibration?  Further, the experiments were quite limited. Studying
the proposed approach in a small, controlled setting might also be
revealing.